# Adenosine Derivates as Antioxidant Agents: Synthesis, Characterization, in Vitro Activity, and Theoretical Insights

**DOI:** 10.3390/antiox8100468

**Published:** 2019-10-09

**Authors:** Francisco Valdes, Nelson Brown, Alejandro Morales-Bayuelo, Luis Prent-Peñaloza, Margarita Gutierrez

**Affiliations:** 1Organic Synthesis Laboratory and Biological Activity (LSO-Act-Bio), PhD Sciences Mention Investigation and Development of Bioactive Products, Institute of Chemistry of Natural Resources, Universidad de Talca, Casilla 747, Talca 3460000, Chile; franciscovaldesz@hotmail.es; 2Center for Medical Research, University of Talca School of Medicine, Talca 3460000, Chile; nbrown@utalca.cl; 3Programa de Investigacion Asociativa en Cancer Gastrico (PIA-CG), Universidad de Talca, Talca 3460000, Chile; 4Centro de Investigación de Procesos del Tecnológico Comfenalco (CIPTEC), Programa de Ingeniería Industrial, Fundación Universitaria Tecnológico Comfenalco–Cartagena, Cr 44 D N 30A, 91, Cartagena-Bolívar 130001, Colombia; alejandromoraba@hotmail.com; 5Organic Synthesis Laboratory and Biological Activity (LSO-Act-Bio), PhD Applied Sciences, Faculty of Engineering and Institute of Chemistry of Natural Resources, Universidad de Talca, Casilla 747, Talca 3460000, Chile; luisprent@gmail.com

**Keywords:** adenosine derivatives, antioxidant, scavenger, Molecular Quantum Similarity index, chemical reactivity index, density functional theory (DFT)

## Abstract

In this work, we present results about the synthesis and the antioxidant properties of seven adenosine derivatives. Four of these compounds were synthesized by substituting the N6-position of adenosine with aliphatic amines, and three were obtained by modification of the ribose ring. All compounds were obtained in pure form using column chromatography, and their structures were elucidated by infrared spectroscopy (IR) and Nuclear Magnetic Resonance (NMR). All adenosine derivatives were further evaluated in vitro as free radical scavengers. Our results show that compounds **1c**, **3**, and **5** display a potent antioxidant effect compared with the reference compound ascorbic acid. In addition, the absorption, distribution, metabolism and excretion (ADME) calculations show favorable pharmacokinetic parameters for the set of compounds analyzed, which guarantees their suitability as potential antioxidant drugs. Furthermore, theoretical analyses using Molecular Quantum Similarity and reactivity indices were performed in order to discriminate the different reactive sites involved in oxidative processes.

## 1. Introduction

During the last decade, antioxidants have become of increasing interest for food, cosmetic, and pharmaceutical applications [1]. Antioxidants are unstable chemical species capable of capturing free radical compounds, which are known to play a significant role in the pathophysiology of a number of disease conditions, including cancer, Alzheimer’s disease, atherosclerosis, hyperuricemia, cellular senescence, cardiovascular conditions, and drug-induced toxicity [2]. Mechanistically, antioxidants can scavenge reactive oxygen species (ROS), inhibit the enzymes responsible for the production of ROS, and/or chelate metals such as iron and copper [3] with the purpose of preventing and/or repairing the damage caused by these radical species [3,4].

Antioxidants can be endogenous molecular complexes such as enzymes or may be incorporated into the body through food and other dietary sources [5]. Among the anti-radical properties of antioxidants, the key lies in their ability to transfer an H-atom of an OH group to free radicals such as hydroxyl (·OH) and superoxide (O_2_·^−^) [6].

Currently, the antioxidant activities of only a few synthetic compounds have been assessed [7]. Therefore, the synthesis of new chemical compounds with antioxidant potential is increasingly necessary [8]. Some synthetic compounds with antioxidant activity comparable to natural antioxidants have been developed and incorporated into food products [9]. Indeed, many of these compounds are extensively used in food products in order to inhibit the process of lipid oxidation. For example, tert-butylhydroquinone (TBHQ), butylated hydroxyanisole (BHA), butylated hydroxytoluene (BHT), and propyl gallate have all been approved by the Food and Drug Administration (FDA) and are now widely added to food products [10]. These antioxidants are cheaper and easier to process than natural antioxidants [11,12].

Adenosine is a natural nucleoside composed of a nitrogenous base of adenine attached to a D-ribose sugar through a β-glycosidic bond (formed between the 1′-position carbon of the pentose and the *N*^9^-position of the purine). This nucleoside plays important roles in several biochemical processes. For example, it functions in reactions of energy transfer as adenosine triphosphate (ATP) and adenosine diphosphate (ADP) as well as in signal transduction pathways as cyclic adenosine monophosphate (cAMP) [13,14]. It also functions as a neuromodulator that is believed to play a role in promoting sleep and suppressing arousal [15] as well as in the regulation of the blood flow to various organs [16]. Of note, this nucleoside is the only known endogenous ligand capable of binding to one of four known adenosine receptors. After binding to these receptors, adenosine regulates a number of physiological processes through differential and cell type-specific activation of these receptors [17,18].

Thus far, compounds derived from adenosine have been synthesized through modifications introduced at the exocyclic amine, the heteroaromatic carbon ring, or the methylene group at the exocyclic ribose ring [19,20]. Among the activities described for adenosine and its semisynthetic derivatives, the antioxidant capacity has also been evaluated [21], showing free radical scavenging activity that is cardioprotective in vivo. In 2012, the synthesis and the biological evaluation of a series of adenosine derivatives were described, demonstrating an ability to reduce ROS production for two of them [22].

Based on these data and our own experience with the synthesis of adenosine derivatives [23], we set out to synthesize a series of novel adenosine derivatives. Following an evaluation of their lipophilicity, these compounds were further tested for their antioxidant capacity using in vitro spectrophotometric analyses [(2,2′-azino-bis (3-ethylbenzothiazoline-6-sulfonic acid) (ABTS) and 1,1-diphenyl-2-picrylhydrazyl(DPPH) assays]. In addition, theoretical studies of molecular quantum similarity measure allowed us to predict biological activities, and local reactivity descriptors such as local softness and electrophilicity indices were obtained with the help of Fukui function calculation. Additionally, in-silico computations of molecular properties, physicochemical profiles, drug scores, and drug-likeness were performed to predict the pharmacokinetic and the toxicity properties (ADME-T) of the biologically active compounds in order to assess their suitability as possible orally-active drug candidates.

## 2. Materials and Methods 

### 2.1. Chemistry Materials

Melting points were calculated using a Büchi apparatus(Stone, Staffs, UK). The progress of reactions and the purity of compounds were checked through analytical thin layer chromatography (TLC) on silica gel plates (Merck 60 F254, KGaA Darmstadt, Germany). Whenever necessary, compounds were purified by column chromatography using mixtures of solvents in crescent polarity. Chemicals were purchased from Aldrich (KGaA Darmstadt, Germany). ^1^H and ^13^C NMR spectra (400 MHz for proton and 100 MHz for carbon) were recorded in an AM-400 spectrometer (Bruker, Rheinstetten, Germany) using DMSO-*d*_6_ and CDCl_3_. Tetramethylsilane (TMS) was used as an internal standard. Chemical shifts (δ) and *J* values are reported in ppm and Hz, respectively, relative to the solvent peak CDCl_3_ at 7.26 ppm for protons and 77 ppm for carbon atoms, and DMSO-*d*_6_ 2.5 ppm for protons and 39.7 ppm for carbon atoms. Signals were designated as follows: s, singlet; d, doublet; dd, doublet of doublets; t, triplet; m, multiplet; br. s, broad singlet. IR spectra (KBr pellets, 500–4000 cm^−1^) were recorded on a NEXUS 670 FT-IR spectrophotometer (Thermo Nicolet, Madison, WI, USA).

### 2.2. Synthetic Procedures

Compounds were prepared by condensation of 6-chloropurine riboside (6-chloro-9-b-d-ribofuranosyl-9*H*-purine) (**1**) with amines containing linear and branched chains as well as aromatic amines by nucleophilic aromatic substitution. Using this approach, we obtained the following compounds:

Compounds **1a**–**c**: these compounds were obtained by nucleophilic attack of the respective amine over the adenosine starting reagent using *N*,*N*-diisopropylethylamine (DIPEA) as a Lewis base (Scheme 1). The nucleoside derivatives were obtained through semisynthetic methods described previously by Ottria et al. in 2010 [24]. In a round-bottomed flask, the 6-chloropurine riboside (0.35 mmol, **1**) was dissolved in absolute EtOH or dimethylformamide (DMF) followed by the addition DIPEA (1.05 mmol) and the appropriate amine (4.5 mmol). The mixture was refluxed at 80 °C with stirring for 8 h. The reaction mixture was cooled down to room temperature. The solvent was removed by filtration (once we had a solid precipitate) or under vacuum to leave a residue that was further analyzed by TLC. The residue was washed with hexane, dried, and purified by SiO_2_ column chromatography (CH_2_Cl_2_-MeOH, 97:3). In some cases, the addition of dry Et_2_O was used to precipitate DIPEACl (*N*,*N*-diisopropylethylamine chloride), which was then filtered off. The crude residue obtained after evaporation was purified by column chromatography.

2-(6-butylamino-purin-9-yl]-5-hydroxymethyl-tetrahydro-furan-3,4-diol (**1a**). Reagents: 6-chloropurine riboside (**1**) and butylamine. Amorphous white solid with 43.17% yield; mp 162–165 °C. ^1^H NMR (DMSO-*d*_6_, 400 MHz); δ: 8.36 (s, 1H, CH-Ar purine); 8.15 (s, 1H, CH-Ar purine);7.83 (s, 1H, NH); 5.90 (d, *J* = 6.11 Hz, 1H, CH-1′); 5.44 (s, 2H, 3′-OH, 5′-OH); 5.17 (s, 1H, 2′-OH); 4.61 (s, 1H, CH-2′); 4.18 (s, 1H, CH-3′); 3.93 (d, *J* = 3.93, 1H, CH-4′); 3.64 (m, 2H, CH_2_-5′);3.40 (s, 2H, CH_2_-R); 1.56 (m, 2H, CH_2_-R); 1.29 (m, 2H, CH_2_-R); 0.87 (t, *J* = 7.15 Hz,3H, CH_3_-R). ^13^C NMR (DMSO-*d*_6_, 100 MHz) δ: 155.16, 152.83, 148.64, 140.16, 120.18, 86.39, 83.43, 73.93, 71.15, 62.14, 31.63, 29.47, 20.03, 14.20. IR (KBr) λ/cm^−1^: 3444, 3414, 2955, 2926, 1630, 1485, 1340, 1218. Anal. Cal. C_14_H_21_N_5_O_4_: C = 51.92%, H = 6.49%, N = 21.63%. Compound **1a** was reported previously [24].

2-[6-(5-amino-2-methyl-pentylamino)-purin-9-yl]-5-hydroxymethyl-tetrahydro-furan-3,4-diol (**1b**). Reagents: 6-chloropurine riboside (**1**) and 1,5-diamine-2-methylpentane. Yellow solid, 48.09% yield; mp 145–148 °C. ^1^H NMR (DMSO-*d*_6_, 400 MHz, δ) 8.32 (m, 1H, CH-Ar purine), 8.17 (d, *J* = 7.83 Hz, 1H, CH-Ar purine), 7.86 (s, 1H, NH); 5.88 (d, *J* = 6.11 Hz, 1H, CH-1′), 5.48 (m, 1H, 2′-OH), 5.45 (m, 1H, 3′-OH), 5.22 (m, 1H, 5′-OH), 4.60 (t, *J* = 5.38 Hz, 1H, CH-2′), 4.14 (pseudo t, *J* = 4.65 Hz, *J* = 3.18 Hz, 1H, CH-3′), 3.95 (q, *J* = 3.18 Hz, 1H, CH-4′), 3.65 (m, 2H, CH_2_-5′), 3.54 (dd, *J* = 11.98 Hz, *J* = 3.42 Hz, 2H, CH_2_-R), 3.15 (s, 2H, CH_2_-R), 1.77 (s, 2H, NH_2_-R), 1.62 (m, 1H, CH-R), 1.15 (m, 2H, CH_2_-R), 0.88 (m, 3H, CH_3_-R). ^13^C NMR (DMSO-*d*_6_, 100 MHz, δ) 152.83, 148.78, 140.53, 140.08, 120.11, 88.35, 86.35, 73.99, 71.10, 62.10, 46.98, 45.18, 31.88, 31.20, 26.60, 17.83. IR (KBr) λ/cm^−1^: 3384, 2921, 2850, 1626, 1466, 1232. Anal. Cal. C_16_H_26_N_6_O_4_: C = 52.40%, H = 7.1%, N = 22.92%.

2-[6-(3-propylamino)-purin-9-yl]-5-hydroxymethyl-tetrahydro-furan-3,4-diol (**1c**). Reagents: 6-chloropurine riboside (**1**) and 1,3-diaminepropane. Yellow solid, 61.47% yield; mp 184–186 °C. ^1^H NMR (DMSO-*d*_6_, 400 MHz, δ) 8.33 (s, 1H, CH-Ar purine); 8.18 (s, 1H, CH-Ar purine); 7.96 (s, 1H, NH), 5.88 (d, *J* = 6.11 Hz, 1H, CH-1′), 4.61 (t, *J* = 5.50 Hz, 1H, CH-2′), 4.16 (dd, *J* = 4.52 Hz, *J* = 3.30 Hz, 1H, CH-3′), 3.96 (m, 1H, CH-4′), 3.66 (m, 1H, CH-5′), 3.56 (d, *J* = 3.67 Hz, 1H, CH-5′), 3.52 (d, *J* = 3.42 Hz, 2H, CH_2_-R), 2.62 (t, *J* = 6.36 Hz, 2H, CH_2_-R), 1.66 (m, 2H, CH_2_-R). ^13^C NMR (DMSO-*d*_6_, 100 MHz, δ) 155.14, 152.83, 148.65, 140.12, 120.22, 88.46, 86.38, 74.04, 71.08, 62.11, 39.27, 38.92, 32.03. IR (KBr) λ/cm^−1^: 3372, 3340, 3146, 2940, 2860, 1629, 1587, 1477, 1297, 1219. Anal. Cal. C_13_H_20_N_6_O_4_: C = 48.10%, H = 6.17%, N = 25.90%.

Compounds **2** to **5**: These derivatives of adenosine were obtained by successive modifications of the commercial precursor 6-chloropurine riboside (**1**) in the ribose ring and by nucleophilic substitution at the *N*^6^-position of the purine ring, protection of vicinal diols 2′-OH and 3′-OH (compound **2**), and then total oxidation of the 5′-OH group to an acid group, forming compound **3**. After the carboxyl group (COOH group) formation, the vicinal diols were deprotected by the action of formic acid (HCOOH) to obtain compound **4**. Immediately, an amidation reaction was carried out to generate an uronamide group at the 5′-carbon position of the furanose ring. This compound was subjected to nucleophilic substitution at the *N*^6^-position of the purine ring with 5-amino-1,3,4-thiadiazole-2-thiol, forming compound **5**. Scheme 2 shows the general procedure and the reaction conditions.

[6-(6-Chloro-purin-9-yl)-2,2-dimethyl-tetrahydro-furo[3,4-d][1,3]dioxol-4-yl]-methanol (**2**): 200 mg (0.7 mmol) of the commercial precursor 6-chloropurine riboside (**1**) and acetone (10 mL) were stirred at room temperature for 30 min, after which *p*-toluensulfonic acid (5.57 mmol) was added. The reaction was stirred at room temperature for 3 h. The reaction progress was monitored by TLC. Sodium bicarbonate (1.5 g) was added, and the mixture was maintained under agitation. Once the reaction was finished, the solid phase was removed by filtration and washed with ethyl acetate (×2). The product was then purified by column chromatography with mixtures of CH_2_Cl_2_-MeOH, obtaining the compound **2**. Yellow solid, 78.1% yield; mp 155–158 °C ^1^H NMR (CDCl_3_, 400 MHz, δ) 8.72 (s, 1H, CH-Ar purine); 8.31 (s, 1H, CH-Ar purine); 6.01 (d, *J* = 8.0 Hz, 1H, CH-1′); 5.16 (m, 1H, CH-2′); 4.97 (d, *J* = 7.83 Hz, 1H, CH-3′); 4.52 (d, *J* = 1.22 Hz, 1H, CH-4′); 3.83 (m, 2H, CH_2_-5′); 5.06 (m, 1 OH); 1.62 (s, 3H, ketal); 1.35 (s, 3H, ketal). ^13^C NMR (CDCl_3_, 100 MHz, δ) 151.6, 151.4, 148.8, 144.4, 132.5, 114.0, 93.4, 86.2, 83.2, 81.1, 62.7, 27.1, 24.8. IR (KBr) λ/cm^−1^ 3320, 2906, 2863, 959, 733. Anal. Cal. C_13_H_15_ClN_4_O_4_: C = 47.75%, H = 4.59%, Cl = 10.85%, N = 17.14%. Compound **2** was reported previously [23].

6-(6-Chloro-purin-9-yl)-2,2-dimethyl-tetrahydro-furo[3,4-d][1,3]dioxole-4-carboxylic acid (**3**): For oxidation of the 5′-OH, 100 mg of compound **2** (0.31 mmol) were dissolved in H_2_O/CH_3_CN 1:1 and placed in an ultrasound bath for 30 min. The solvent was removed by vacuum, and the residue obtained was stirred with diethyl ether (50 mL), filtered, and then dried before being purified in a SiO_2_ column chromatography with mixtures of CH_2_Cl_2_-MeOH, obtaining the compound **3**. Yellow solid, 83.5% yield; mp 209–211 °C. ^1^H NMR (DMSO-*d*_6_, 400 MHz, δ) 9.20 (s, 1H, CH-Ar purine); 8.73 (s, 1H, CH-Ar purine); 6.31 (s, 1H, CH-1′); 5.28 (d, *J* = 5.62, 1H, CH-2′); 5.20 (d, *J* = 5.62 Hz, 1H, CH-3′); 4.54 (s, 1H, CH-4′); 1.52 (s, 3 H, ketal); 1.31 (s, 3H, ketal). ^13^C NMR (DMSO-*d*_6_, 100 MHz) δ: 173.1, 151.9, 150.9, 149.1, 147.3, 131.4, 112.8, 91.7, 88.2, 84.9, 84.4, 27.2, 25.5. IR (KBr) λ/cm^−1^ 3409, 2990, 2937, 1592, 1336, 1206, 1086, 635. Anal. Cal. C_13_H_13_ClN_4_O_5_: C = 45.79%, H = 3.82%, Cl = 10.40%, N = 16.44%. The compound **3** was reported previously [23].

1’-deoxy-1’-(6-chloro-9H-purin-9-yl)-β-d-ribofuranuronic acid (**4**): The deprotection of diols present at the 2′ and the 3′ position of the ribose was carried out by an acid hydrolysis of the cyclic ketal group. For this reaction, compound **3** was mixed in 50% formic acid (HCOOH) at 80 °C, and the product was washed and dried under vacuum. Then, the compound was purified, and a brown amorphous solid was formed with a yield of 53.02%. mp 135–140 °C. ^1^H NMR (DMSO-*d*_6_, 400 MHz, δ) 12.40 (s, 1H, –COOH); 8.38 (d, *J* = 13.94 Hz, 1H, CH-Ar purine); 8.04 (d, *J* = 16.00 Hz, 1H, CH-Ar purine); 6.02 (d, *J* = 6.36 Hz, 1H, CH-1′); 4.55 (m, 1H, CH-2′); 4.48 (m, 1H, CH-3′); 4.33 (s, 1H, CH-4′). ^13^C-NMR (DMSO-*d*_6_, 100 MHz, δ:) 171.23, 156.95, 155.33, 146.61, 146.10, 138.79, 87.57, 82.94, 74.34, 73.60. IR (KBr) λ/cm^−1^ 3492, 2956, 2867, 1683, 1206, 607. Anal. Cal. C_10_H_9_ClN_4_O_5_: C = 39.91%, H = 2.99%, Cl = 11.79%, N = 18.63%.

6-[(4-amino-2,3,5-triazole) thio]-β-d-ribofuranosyl-9*H*-purine-5′-*N*-ethyluronamide (**5**): Compound **5** was synthesized in a reaction involving compound **3**, 1-ethyl-3-(3-dimethylaminopropyl) carbodiimide (EDC), and hydroxybenzotriazole (HBOt). Ethylamine (EtNH_2_) was immediately added, and the mixture was left stirring for 24 h at room temperature in DMF. Subsequently, *N*,*N*-diisopropylethylamine (DIPEA) and 5-amino-1,3,4-thiadiazole-2-thiol were added in situ to carry out the aromatic nucleophilic substitution reaction. The reaction was refluxed at 80 °C and under stirring in DMF for 8 h according to the modified procedures of Ottria et al., 2010 [24]. The resulting product was purified and dried, forming a brown solid with 47% yield. mp 135–138 °C. ^1^H NMR (DMSO-*d*_6_, 400 MHz, δ) 8.97 (s, 1H, CH-Ar purine), 8.59 (s,1H, CH-Ar purine), 7.80 (s, 1H, NH), 7.56 (s, 2H, NH_2_); 6.20 (s, 1H, CH-1′), 5.21 (d, *J* = 5.87 Hz, 1H, CH-2′), 5.07 (d, *J* = 5.87 Hz, 1H, CH-3′), 4.38 (s, 1H, CH-4′), 2.70 (m, 2H, CH-5′), 1.42 (s, 3H, CH_3_-cetal), 1.21 (s, 3H, CH_3_-cetal), 1.00 (d, *J* = 6.60, 3H, CH_3_-R). ^13^C NMR (DMSO-*d*_6_, 100 MHz, δ) 173.56, 162.79, 155.65, 151.95, 149.58, 145.97, 141.10, 130.75, 112.79, 91.34, 88.14, 84.86, 84.51, 36.26, 27.27, 25.55, 18.78. IR (KBr) λ/cm^−1^ 3430, 3146, 2984, 2932, 1633, 1560, 1494, 1074 cm^−1^. Anal. Cal. C_17_H_20_N_8_O_4_S_2_: C = 43.92%, H = 4.31%, N = 24.11%, O = 13.78%, S = 13.78%.

### 2.3. Antioxidant Activity

All derivatives of adenosine were evaluated as potential antioxidant agents by analyzing free radical scavenging of DPPH and ABTS.

#### 2.3.1. DPPH Assay

Measurement of DPPH radical scavenging activity—DPPH is a stable free radical capable of accepting an electron or a hydrogen radical. The antioxidant action is reflected by a change in coloration from deep violet to yellow after adding the compounds to a methanolic solution of DPPH. The free radical scavenging effect of the compounds was assessed by the discoloration of a methanolic solution of DPPH, as previously reported [25]. Adenosine derivatives were tested at 100, 50, and 10 μg/mL. Scavenging of free radicals by all adenosine derivatives was evaluated spectrophotometrically at 517 nm against the absorbance of the DPPH radical. The percentage of discoloration was calculated as follows:% scavenging DPPH free radical = 100 × (1 − AE/AD)
where AE is the absorbance of the solution after adding the extract, and AD is the absorbance of the blank DPPH solution. Ascorbic acid was used as a reference compound with IC_50_ value of 1.5 μg/mL.

#### 2.3.2. ABTS Assay

Measurement of ABTS radical scavenging activity—ABTS (2,2′-azino-bis (3-ethylbenzothiazoline-6-sulfonic acid) radical scavenging assay is a method based on the capacity of compounds with antioxidant activity to donate hydrogen and scavenge the long-lived radical cation ABTS·^+^. In this method, the preformed radical mono-cation of ABTS is first generated by oxidation of ABTS with potassium persulfate and is subsequently reduced in the presence of hydrogen donating antioxidants [26]. The ABTS assay was performed according to a protocol previously described [7,27]. The ABTS radical cation (ABTS·^+^) was formed in a reaction of 7 mM of ABTS with 2.45 mM potassium persulfate. The mixture was kept in the dark at room temperature for 12 h before use. The ABTS·^+^ solution was then diluted with ethanol to give an absorbance of 0.7 ± 0.01 at 745 nm. Under these conditions, a solution of a test compound was allowed to react with the ABTS·^+^ solution (proportion 1:2), and the absorbance was measured at 745 nm after 1 min. Adenosine derivatives were tested at concentrations of 100, 50, and 10 μg/mL. Data for each assay were recorded in triplicate. Ascorbic acid was used as positive control with an IC_50_ of 27.62 μg/mL. The scavenging activity was calculated based on the percentage of ABTS radicals scavenged by the formula:

% scavenging = [(A_0_ − As)/A_0_] × 100

where A_0_ is absorption of control, and A_S_ is absorption of the tested compound solution.

### 2.4. Computational Methods

#### 2.4.1. In Silico Prediction of Pharmacokinetic Properties

The pharmacokinetic properties of the adenosine derivatives were calculated in silico through absorption, distribution, metabolism and excretion (ADME) descriptors using QikProp [28], which was based on Lipinski’s rule of five. The descriptors were molecular weight, Van der Waals interactions, surface areas of polar nitrogen and oxygen atoms, H bond acceptors, H bond donors, Log P (octanol/water), and aqueous solubility.

#### 2.4.2. Molecular Quantum Similarity Measures

Molecular quantum similarity measurement (MQSM) between two systems A and B, denoted ZAB, is a comparison between two molecules that can be constructed using their respective density functions (DFs). Both DFs can be multiplied and integrated over all the respective electronic coordinates and, in turn, weighed by a defined positive operator Ω (r1, r2) [29,30,31].
(1)$$Z_{AB}=\langle \rho A|\Omega |\rho B\rangle =\int \int \rho A(r1)\Omega (r1,r2)\rho B(r2)dr1dr2$$

The nature of the operator used in Equation (1) provides the information to be compared between the two systems.

#### 2.4.3. Molecular Similarity Indexes

Once a group of studied objects along with the operator derived from the Molecular Quantum Similary Measures (MQSM) [Equation (1)] are chosen, the measure of similarity obtained for the group is unique; however, it is common practice to transform or combine these measures to obtain new classes of auxiliary terms that can be called quantum similarity indices (QSI). There is a vast amount of possible manipulations of MQSM leading to a variety of QSI definitions. The following are the most commonly used definitions and, for this reason, were the ones used in this work [31,32,33,34]:

Carbó’s similarity index between two molecules, I and J, is generated according to the following Equation (2):(2)$$C_{IJ}\left(\Omega \right)=\;z_{IJ}\left(\Omega \right){\left[z_{II}\left(\Omega \right)z_{JJ}\left(\Omega \right)\right]}^{-1/2}$$

The equation corresponds to the cosine of the angle subtended by the density functions involved, taken as vectors. For this reason, the index is also called the cosine-like similarity index. This Carbo’s QSI, for any pair of molecules, has a value between 0 and 1, depending on the similarity associated with the two molecules (when I = J, the index approaches 1) [31,32,33,34,35].

The quantum similarity index using the Euclidean distance considers Equation (3).
(3)$$D_{IJ}\left(k,x,\Omega \right)={\left[k\left(z_{II}\left(\Omega \right)+z_{JJ}\left(\Omega \right)\right)/2\;-\;xz_{IJ}\left(\Omega \right)\right]}^{1/2},\;x\left[0,\;k\right]$$

For this equation, if k = x = 2, the QSI is reduced to the so-called Euclidean distance index. We can also define the index 3 as:(4)$$D_{IJ}\left(\infty ,\Omega \right)=\;max\left(z_{II}\left(\Omega \right),\;z_{JJ}\left(\Omega \right)\right)$$

This Equation (4) constitutes the distance index of infinite order [30].

#### 2.4.4. Types of Measures in Molecular Quantum Similarity

It depends essentially on the information provided, particularly on the selection of the supported operators producing different types of MSQM [36,37].

Overlap MQS, considering Equation (2). It is the simplest and the most intuitive choice for a positively defined operator. It is the distribution Dirac’s delta, Ω (r1, r2) = δ (r1, r2). This selection transforms the general definition of MQSM in order to calculate the overlap MQSM, which provides measurements of the volume enclosed in the superposition of both electronic density functions [35,36,37].
(5)$$z_{IJ}(\Omega )=\int \int \rho_{I}(r_{1})\delta (r_{1}-r_{2})\rho_{J}(r_{2})dr_{1}dr_{2}=\int \rho_{I}(r)\rho_{J}(r)dr$$

The Dirac’s delta function arises intuitively from its physical definition and is computationally compliant. The MQSM arises from the information on the concentration of electrons in a molecule and indicates the degree of overlap between molecular comparisons [34,35,36,37,38]. Coulomb MQS, considering Equation (2):

If the operator (Ω) is adopted by the Coulomb operator, Ω (r1, r2) = $\frac{1}{\left|r_{1}-r_{2}\right|}$, it provides the Coulomb MQS, which represents the electrostatic repellent Coulomb energy between two charge densities [37,38]:(6)$$Z_{IJ}(\Omega )=\int \int \rho_{I}(r_{1})\frac{1}{r_{1}-r_{2}}\rho_{J}(r_{2})dr_{1}dr_{2}$$

The Coulomb operator has an effect on the overlap density functions. Considering the functions of molecular density as an electron distribution in space, this expression is only for the extension of Coulomb for the distribution of continuous charge. For that reason, it can be considered—in some occasions—as a descriptor of electrostatic potential. This operator provides a measurement of electrostatic repulsion between electronic distributions and is associated with electrostatic interactions [33,34,35,36].

Euclidean distance index, considering Equation (3). This is another typical transformation that can be defined according to the classical distance:(7)dab=[∑j=1p(Δxj)k]1k
where *d_ab_* is the distance between the objects *a* and *b*, *k* = 2, for the definition of the distance. The Euclidean distance between two quantum objects A and B is defined by the following mathematical expression [35,36,37,38]:(8)$$d_{ab}=\sqrt{{\left(x_{a}-x_{b}\right)}^{2}}$$

Occasionally, it is expressed as $D_{AB}=\sqrt{Z_{AA}+Z_{BB}+ZZ_{AB}}$ and has values in the range of [0, ∞) but converges for previous cases, and it has a value of zero between the compared objects if the compared objects are identical [35,36,37,38]:(9)$$D_{AB}=0$$

Geometrically, this index can be interpreted by the norm of the differences between the density functions of the compared objects. The index of the Euclidean distance can be defined by the distance or the dissimilarity index, and the index can also be expressed as [33,34,35]:(10)$$D_{AB}=∥\rho_{A}-\rho_{B}∥=\sqrt{{\left(\rho_{A}-\rho_{B}\right)}^{2}}$$

#### 2.4.5. Alignment Method: Topo-Geometrical Superposition Algorithm (TGSA)

In this work, alignments were performed using the Topo-Geometrical Superposition Algorithm (TGSA) method [39]. This method considers that the optimal alignment of molecules occurs through superposition on a common skeleton. It only takes into account the type of atoms and the bonds formed by interatomic interactions. To carry out its purpose, the algorithm examines the atomic pairs of the molecules and aligns the common substructure for a series of molecules [39]. The method is only based on topology and geometric considerations, where the molecular topology is manifested in the way of comparing the distant bonds. For two molecules, the superposition is unique and does not depend on the type of operator chosen to provide the meaning of similarity [39].

#### 2.4.6. DFT-Based Reactivity Descriptors

Some researchers have shown a relationship between quantum similarity and chemical reactivity descriptors [40,41,42,43,44,45,46,47]. In addition, quantum similarity and DFT use the density function as the object of study, and the similarity indexes, specifically the Coulomb index, can be related to electronic factors associated with chemical reactivity.

Using frontier molecular orbitals (FMO) and the energy gap, global reactivity indices such as chemical potential (μ) [48], hardness (η) [49], and electrophilicity (ω) [48,49,50] can be calculated. These chemical reactivity indices provide an idea about the stability of a system.

The chemical potential (μ) characterizes the tendency of electrons to escape from the equilibrium system [48], whereas the chemical hardness (η) is a measure of the resistance of a chemical species to change its electronic configuration [49].
(11)$$\mu \approx \frac{E_{LUMO}+E_{HOMO}}{2}$$
(12)$$\eta \approx E_{LUMO}-E_{HOMO}$$

The electrophilicity index (ω) can be interpreted as a measure of the stabilization energy of the system when it is saturated by electrons from the external environment and is mathematically defined as [49,50]:
*ω* = *μ*^2^/*2**η*(13)

In this work, the local reactivity descriptor is based on Fukui functions [Equations (14) and (15)]. Equations (14) and (15) represent the response of the chemical potential of a system to changes in the external potential. It is defined as the derivative of the electronic density with respect to the number of electrons at constant external potential:(14)$$f_{k}^{+}\approx \int_{k}\left[\rho_{N+1}\left(\vec{r}\right)-\rho_{N}\left(\vec{r}\right)\right]=\left[q_{k}\left(N+1\right)-q_{k}\left(N\right)\right]$$
(15)$$f_{k}^{-}\approx \int_{k}\left[\rho_{N}\left(\vec{r}\right)-\rho_{N-1}\left(\vec{r}\right)\right]=\left[q_{k}\left(N\right)-q_{k}\left(N-1\right)\right]$$
where ($f_{k}^{+}$) is for nucleophilic attack and ($f_{k}^{-}$) for electrophilic attack [50,51,52,53,54]. In this sense, using the global and the local reactivity descriptors, it is possible to study the quantum dissimilarity along a molecular set.

## 3. Results and Discussion

### 3.1. Synthesis and Characterization of Adenosine Derivatives

Seven adenosine derivatives were synthetized from the compound 6-chloropurine riboside (6-chloro-9-b-d-ribofuranosyl-9*H*-purine). Four derivatives were synthesized by substituting the *N*_6_-position with aliphatic amines, and three were obtained by modification of the ribose ring. All compounds were purified by column chromatography. The compounds were characterized by IR and NMR spectroscopy, showing the presence of functional groups with signals at 3500 cm^−1^ typical of hydroxyl groups, bands between 1640 and 1560 cm^−1^ for primary amines, and bands around 1500 cm^−1^ for secondary amines. ^1^H NMR spectra showed signals characteristic of aromatics protons near to heteroatoms and aliphatic protons present in the compounds. IR as well as ^1^H and ^13^C NMR spectra of all synthetic compounds are included in the Appendix A.

The structural diversity of the six synthetic compounds is presented in Figure 1. Derivatives **1a**, **1b**, and **1c** were obtained by nucleophilic aromatic substitution (SNAr) at the *N*^6^-position of the chloropurine riboside, and compounds **2**, **3**, and **5** were obtained by modification of the ribose ring, protection of the vicinal diols 2′-OH and 3′-OH [55], and/or total oxidation of the 5′-OH group [56]. In the particular case of compound **5**, following the modification in the ribose ring, an uronamide group at the 5′-position of the pentose was formed, and the substitution was carried out between the chlorine atom of the nucleoside (6-position of purine ring) and the thiol group of the amine. This was because of the higher reactivity and nucleophilia of the thiol group compared to the amine group contained in the same compound.

Compound **4** is a derivative of 6-chloropurine riboside that was subjected to total oxidation at the 5′-OH group. For biological assays, both the pure adenosine and the 6-chloropurine riboside were included as controls (Figure 2).

### 3.2. Scavenging Activity of the DPPH Free Radical

Overall, the aim of radical scavenging experiments is to provide information regarding the potential antioxidant capabilities of compounds based on the relationship between structure and antioxidant activity. All adenosine derivatives were tested as antioxidant agents in DPPH (1,1-diphenyl-2-picrylhydrazyl) assays. The antioxidant values of these compounds were then compared with the values obtained for the precursor adenosine as well as for the positive control ascorbic acid. As shown in Table 1, the values of DPPH scavenging activity varied widely, departing from the values obtained for the reference inhibitor. The most active compounds were **5** and **1c**; however, the IC_50_ values were not comparable with those of the control.

### 3.3. Scavenging Activity of the Free Radical ABTS

As shown in Table 2, compounds **2**, **5**, and **1a** displayed excellent activity in the ABTS radical capture assay with IC_50_ values better than those obtained with ascorbic acid. Of note, while compound **2** showed no differences with the precursor in terms of antioxidant activity, compounds **1c** and **5** showed high activity compared with ascorbic acid. These activities can be related to the patterns of substitution at the *N*^6^ position with NH_2_ groups showing an inducing effect by acting as electron donating groups.

### 3.4. Computational Results

To examine potential biological activities, analyses of chemical reactivity were performed. To this end, quantum similarity fields and chemical reactivity frameworks were used. Taking into account the structural features of these molecular sets, which have only one substitute, the molecular quantum similarity indices allowed us to quantify the structural and the electronic effects from a local point of view.

Shown in Table 3 are the similarity values using the overlap operator. These measures relate the structural details and the steric effects of a molecular set. The highest value (0.9856) with a Euclidean distance of 0.7458 (see Table 4) was obtained between adenosine and the 6-chloropurine riboside (**1**). These compounds have an amine and a chloride group, respectively. The lowest value (0.2033) with a Euclidean distance of 7.0771 (see Table 4) was obtained between compounds **5** and **1c**. These compounds have high steric effects on their substituent groups.

Table 5 shows the quantum similarity indices calculated using the Coulomb operator. These measures are associated with electronic effects along the molecular set. The highest value (0.9993) with a Euclidean distance of 1.5508 (see Table 6) was obtained between adenosine and the 6-chloropurine riboside. Its value is in agreement with the overlap similarity (see Table 3). The lowest value (0.7597) with a Euclidean distance of 47.2268 (see Table 6) was observed between compounds **5** and **1c**. In general, while several compounds were structurally dissimilar, they were similar from the electronic point of view.

Conceptual DFT reactivity descriptors such as chemical hardness (η), electronic chemical potential (μ), electrophilicity index (ω), softness (S), and Fukui functions (fk^+^ and fk¯) are very helpful for explaining the reactivity of any molecule. The values of the most important reactivity descriptors for all compounds are shown in Table 7. These values can help us to understand the experimental activity reported for these compounds.

According to Table 7, the compound with the highest chemical potential (μ = −4.8028 eV) and hardness (η = 7.5702 eV) is compound **4**. This compound has a softness (S) value of 0.1320 eV^−1^, electrophilicity (ω) of 1.5235 eV, and experimental activity (IC_50_) of more or equal to 100 μg/mL. The compound with the lowest chemical reactivity is **1b** (μ = −3.3141 eV). On the other hand, the compound with the highest electrophilicity is compound **4** (ω = 1.5235 eV). These values of electrophilicity allow us to relate the non-covalent activity in the active site and help us to understand their stabilization on the active site.

In Figure 3, we can see the local reactivity descriptors using the Fukui functions for selected compounds, taking into account experimental activities. In these analyses, the Fukui HOMO and LUMO functions have the same zones on the central ring. Theses insights allow us to understand the retro-donor processes of these compounds, showing another variable in the stabilization at the active site.

### 3.5. In Silico Evaluation of the ADME-T and Drug-Likeness

Drug-likeness describes an integrated equilibrium between multiple molecular properties and structural features that define whether a particular compound is comparable to already known drugs. Among common principles applied to evaluate the drug-like properties of a compound, Lipinski’s rule of five (RO5) and Veber’s criteria are prominent [57]. These properties comprise hydrophobicity, electronic distribution, hydrogen-bonding capability, molecular size, and flexibility, all of which would affect the behavior of a molecule in a living system, including bioavailability, transport, affinity to proteins, reactivity, toxicity, and metabolic stability.

Computational methods constitute important tools that help us to predict some properties of compounds with potential biological activity. For example, QikProp [58] is a quick, accurate, and easy to use computation program that predicts absorption, distribution, metabolism, and excretion (ADME) of compounds [58]. The main parameters on which QikProp is based are shown in Table 8. Considering the Lipinski’s rule of five [molecular weight below 500 Da, less than five hydrogen bond donors and less than 10 hydrogen bond acceptors, Log P (octanol/water) partition coefficient for the ligand of less than five], all synthesized compounds were within the permissible range for each descriptor [59]. Likewise, adenosine derivatives satisfied other parameters involved in absorption, distribution, and membrane penetration, such as predicted skin permeability (Log Kp) [60,61], water solubility (Log S) [62], and polar surface area (PSA) [63]. Finally, the predicted oral absorption [64] was calculated. This prediction was made through the analysis of the adequate values of different descriptors.

Table 8 reports ADME analyses and shows that all compounds have a high probability of not causing cell damage in experimental assays. In general, the new adenosine derivatives display permissible values for different descriptors [65].

## 4. Conclusions

In summary, the synthesis of seven adenosine derivatives was carried out in mild conditions. Four of them were synthesized by substituting the *N*^6^-position with aliphatic amines, and three were obtained by modification of the ribose ring. The compounds were isolated and purified by column chromatography. Their structures were elucidated by IR and NMR. The antioxidant activity was dependent on the concentration of the compounds; likewise, compounds **3**, **5**, and **1c** presented the most favorable antioxidant activity values. The theoretical physicochemical descriptors revealed that the adenosine derivatives had low toxicity risk. Based on their reactivity and their biological activity, these adenosine derivatives are therefore attractive candidates for the production of a new generation of compounds.

Thus, we describe compounds with better antioxidant action than the reference compound. From theoretical calculations (MQSM, global reactivity descriptors, and Fukui functions), we could establish similarities and discriminate different reactive sites in the new molecules where the oxidative process might occur. These sites can be used for the design of new compounds with interesting biological activities. Further studies are in progress in our research group using other in vitro and in silico analyses in order to design, at a molecular level, efficient antioxidant compounds.

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
