# Peer review of "Adenosine Derivates as Antioxidant Agents: Synthesis, Characterization, in Vitro Activity, and Theoretical Insights"

_antioxidants, 2019, doi:10.3390/antiox8100468_

Round 1

Reviewer 1 Report

Dear Editor,

The manuscript submitted by Valdes et al. aims to describe adenosine derivatives for antioxydant properties. This paper is well written. (1) The introduction provide sufficient background and include all relevant references for description, (2) The research design is well appropriated and described, (3) All material & methods are adequately described in the manuscript, (4) all the results are clearly presented and discussed in the manuscript and, (5) The conclusions are supported by the results.

For all these reason, this manuscript could be accepted for publication after minor revisions such as : improving quality of some figures, and revise English.

Regards,

Author Response

Dear Editor and Reviews:

I appreciate the review made, the requested changes have been incorporated: language revision and edition of paragraphs to reduce the similarities detailed, the figures are in the format required by the journal. I appreciate very much you can review the work again and verify the incorporation of the improvements.

Best Regards

 Margarita Gutierrez

Reviewer 2 Report

This work by Francisco Valdes et al. described “Adenosine derivates as antioxidant agents: synthesis, characterization, in vitro activity and theoretical insights”. The paper is complete and well structured. The description of experimental procedure is detailed and results critically analyzed and commented. Particularly interesting are results obtained for compounds 1c, 3 and 5 that have a potent antioxidant effect comparing with the ascorbic acid, and also theoretical studies using Molecular Quantum Similarity Index.

I think the manuscript can be accepted in this form and it needs no revision.

Author Response

(The authors gave the same response as above.)
